# The interplay between asymmetric and symmetric DNA loop extrusion

Edward J Banigan*, Leonid A Mirny*

Department of Physics and Institute for Medical Engineering and Science, Massachusetts Institute of Technology, Cambridge, United States

**Abstract** Chromosome compaction is essential for reliable transmission of genetic information. Experiments suggest that ~1000-fold compaction is driven by condensin complexes that extrude chromatin loops, by progressively collecting chromatin fiber from one or both sides of the complex to form a growing loop. Theory indicates that *symmetric* two-sided loop extrusion can achieve such compaction, but recent single-molecule studies (Golfier et al., 2020) observed diverse dynamics of condensins that perform one-sided, symmetric two-sided, and *asymmetric* two-sided extrusion. We use simulations and theory to determine how these molecular properties lead to chromosome compaction. High compaction can be achieved if even a small fraction of condensins have two essential properties: a long residence time and the ability to perform two-sided (not necessarily symmetric) extrusion. In mixtures of condensins I and II, coupling two-sided extrusion and stable chromatin binding by condensin II promotes compaction. These results provide missing connections between single-molecule observations and chromosome-scale organization.

## Introduction

During mitosis in metazoan cells, each chromosome is linearly compacted ~1000-fold from a ~1-mm-long chromatin polymer globule into a ~1 μm array of chromatin loops (*Paulson and Laemmli, 1977*; *Marsden and Laemmli, 1979*; *Earnshaw and Laemmli, 1983*; *Maeshima et al., 2005*). This remarkable reorganization is primarily driven by the protein complex condensin (*Hirano and Mitchison, 1994*; *Saitoh et al., 1994*; *Saka et al., 1994*; *Strunnikov et al., 1995*; *Hirano et al., 1997*; *Shintomi et al., 2015*; *Gibcus et al., 2018*), which is one of a class of DNA-binding complexes known as structural maintenance of chromosomes (SMC) complexes. Condensin is believed to compact the chromatin fiber by an active process known as 'loop extrusion' (*Yatskevich et al., 2019*; *Banigan and Mirny, 2020*). In the loop extrusion model, a loop-extruding factor (LEF), such as a condensin motor, binds the chromosome and progressively grows a DNA/chromatin loop by translocating along and processively collecting the nearby chromatin fiber (*Riggs, 1990*; *Alipour and Marko, 2012*; *Goloborodko et al., 2016b*). DNA loop extrusion by condensins (*Ganji et al., 2018*; *Golfier et al., 2020*; *Kong et al., 2020*; *Kim et al., 2020*) and other SMC complexes (*Kim et al., 2019*; *Davidson et al., 2019*; *Golfier et al., 2020*) has recently been observed in single-molecule experiments in vitro. However, it has not been established how condensins with the properties observed in vitro can attain the high degree of linear compaction required for mitotic chromosome compaction in vivo.

In the first single-molecule experiments that directly imaged loop extrusion, yeast condensins were observed to extrude DNA loops in an asymmetric, 'one-sided' manner (*Ganji et al., 2018*). In this mode of loop extrusion, part of the condensin complex remains anchored to DNA (*Kschonsak et al., 2017*), while condensin extrudes DNA from one side of the complex into a loop (i.e., collecting DNA either upstream or downstream of the bound site, but not both) (*Ganji et al., 2018*). Importantly, this contrasts with most models for loop extrusion by condensin, in which each loop-extruding factor performs symmetric 'two-sided' extrusion, growing loops by gathering DNA/

*For correspondence:
ebanigan@mit.edu (EJB);
lmirny@gmail.com (LAM)

Competing interests: The authors declare that no competing interests exist.

chromatin from both sides of the protein complex (*Alipour and Marko, 2012*; *Goloborodko et al., 2016b*; *Fudenberg et al., 2017*; *Banigan and Mirny, 2020*). Furthermore, theoretical arguments and computational modeling predict that the observed one-sided activity is insufficient to generate the 1000-fold linear compaction expected for metazoan mitotic chromosomes (*Banigan and Mirny, 2019*; *Banigan et al., 2020*; *Banigan and Mirny, 2020*).

Subsequent single-molecule experiments with condensins revealed different and more diverse properties for loop extrusion, particularly in higher eukaryotes. Recent experiments show that condensins from human and Xenopus cells can perform both one-sided and two-sided loop extrusion (*Golfier et al., 2020*; *Kong et al., 2020*). However, two-sided extrusion by *Xenopus* condensins proceeds *asymmetrically* rather than symmetrically (*Golfier et al., 2020*). Chromosome compaction by condensins performing asymmetric but two-sided loop extrusion has not yet been systematically and quantitatively investigated.

An additional complication is that higher eukaryotes have two types of condensin, condensins I and II (*Ono et al., 2003*), each of which has different properties, including residence times and extrusion speeds (*Gerlich et al., 2006a*; *Walther et al., 2018*; *Kong et al., 2020*). Each of these condensins plays a role in mitotic chromosome compaction (*Ono et al., 2003*; *Hirota et al., 2004*; *Ono et al., 2004*; *Green et al., 2012*; *Ono et al., 2017*; *Gibcus et al., 2018*; *Hirano, 2016*; *Kalitsis et al., 2017*; *Takahashi and Hirota, 2019*), but the linear compaction abilities of mixtures of loop-extruding condensins with different dynamic properties has not been systematically explored.

Using simulations and theory, we investigated whether asymmetric two-sided extrusion or a mixture of one- and two-sided loop-extruding factors (LEFs) with different dynamics can generate the high degree of linear compaction observed for mitotic chromosomes in vivo. We find that asymmetric two-sided extrusion can eliminate unlooped gaps between neighboring LEFs and compact chromosomes >1000-fold. Importantly, compaction can be achieved even with the relatively large asymmetries that are observed in vitro, provided that LEFs are two-sided. We also perform simulations and develop a theory that show that mixtures of one- and two-sided LEFs can achieve high levels of compaction, provided that the two-sided LEFs have sufficiently long residence times. Furthermore, the simulations suggest that mitotic chromosome compaction may require a tight coupling between stable chromatin binding and two-sided extrusion by condensin II complexes, while condensins I and II that dynamically exchange may perform one-sided extrusion. This result suggests that condensin II complexes may dimerize in vivo to promote chromosome compaction. These models provide the first demonstration of how loop-extruding condensins with the properties observed in single-molecule experiments could generate the linear compaction required to form metazoan mitotic chromosomes.

## Model

In the model, LEFs representing SMC complexes perform loop extrusion on a polymer fiber representing the chromosome (*Alipour and Marko, 2012*; *Goloborodko et al., 2016b*; *Fudenberg et al., 2016*; *Banigan et al., 2020*). Each LEF is composed of two subunits or 'sides', which may have different translocation abilities, so the entire LEF may be either symmetric or asymmetric. A subunit may be either active or inactive. An active subunit processively translocates at speed $v$ along the polymer fiber, thus creating and enlarging the polymer loop between the subunits. In this work, inactive subunits are immobile. We refer to LEFs with two active subunits as 'two-sided'. LEFs with one active subunit and one inactive subunit are referred to as 'one-sided'. Each LEF subunit is assumed to act as a barrier to the translocation of other LEFs so that an active subunit cannot pass through another LEF subunit. Thus, pseudoknots or 'Z-loops' (*Kim et al., 2020*) are prohibited; the scenario in which one-sided LEFs may traverse each other has been considered previously (*Banigan et al., 2020*) (moreover, compaction by mixtures of one- and two-sided LEFs in that model would simply rescale the mean processivity compared to the pure one-sided LEF model). LEFs bind with equal probability to any location on the polymer fiber, following a previous analysis of condensin localization that found condensin loading to be largely sequence-independent (*Gibcus et al., 2018*). Furthermore, each one-sided LEF has two possible binding orientations (← or →), which determines the direction in which extrusion proceeds along the

polymer. Following in vitro experiments (*Ganji et al., 2018*; *Golfier et al., 2020*) and lacking molecular evidence that binding to chromatin in a particular orientation could be favored, the extrusion orientation for each LEF is selected randomly. Each LEF stochastically unbinds at rate $k$, which releases the corresponding polymer loop. Altogether, these LEF dynamics lead to a dynamic statistical steady state in which loops formed by LEFs stochastically appear, grow, and vanish. The steady-state fold linear compaction, $\mathcal{FC}$, is calculated from the fraction, $f$, of the fiber that is extruded into loops as $\mathcal{FC} = (1 - f)^{-1}$ (*Banigan and Mirny, 2019*). This quantity primarily depends on the ratio, $\lambda/d$, of the processivity ($\lambda = pv/k$, where $p = 1$ or 2 for one- or two-sided LEFs, respectively, and in mixtures, $\lambda$ denotes the population-averaged processivity) to the mean separation ($d$) between LEFs (*Goloborodko et al., 2016b*; *Banigan and Mirny, 2019*). Further details and a public link to the simulation code are provided in the Materials and methods section.

Previous computational models of loop extrusion generally assumed that all LEFs have the same average unbinding rate and that all active LEF subunits translocate at the same average speed (*Alipour and Marko, 2012*; *Sanborn et al., 2015*; *Goloborodko et al., 2016b*; *Fudenberg et al., 2016*; *Miermans and Broedersz, 2018*; *Banigan and Mirny, 2019*; *Banigan et al., 2020*). Since experiments observed SMC complex dynamics that are contrary to these assumptions, we consider models in which these assumptions are relaxed. In the Results section, we first present results for systems with LEFs that perform asymmetric two-sided loop extrusion. Second, we investigate mixtures of one- and two-sided LEFs with different extrusion velocities, mixtures of one- and two-sided LEFs with different residence times, and mixtures of only one-sided LEFs with different residence times. Third, we specifically consider mixtures of LEFs with the properties measured for condensins I and II in experiments. Additionally, the simulation results for mixtures of one- and two-sided LEFs are explained by theoretical arguments, which are presented in detail in Appendix 1.

## Results

### Asymmetric two-sided extrusion can linearly compact mitotic chromosomes

To determine the ability of asymmetric two-sided loop-extruding condensins (*Golfier et al., 2020*) to compact mitotic chromosomes, we simulated an asymmetric variant of the two-sided loop extrusion model. In these simulations, each LEF has two active subunits. One of these is a fast subunit that extrudes at speed $v_{\text{fast}}$, while the other is a slow subunit that extrudes at speed $v_{\text{slow}}$. LEFs unbind from the chromatin polymer fiber at rate $k = 1/\tau$, where $\tau$ is the mean residence time. Thus, the mean processivity (i.e. how large a loop a LEF can extrude before unbinding) is $\lambda = (v_{\text{fast}} + v_{\text{slow}})\tau$. A prototypical trajectory, a schematic illustration, and an arch diagram are shown in *Figure 1a*.

Eliminating or avoiding unlooped gaps between LEFs is critical to achieving a high degree of linear compaction (*Banigan and Mirny, 2019*; *Banigan et al., 2020*). One-sided LEFs cannot close all gaps because one quarter of all pairs of neighboring LEFs are in a divergent orientation ($\leftarrow\rightarrow$); thus, they extrude loops by collecting chromatin from opposite directions, while leaving the chromatin between the LEFs unlooped (*Banigan and Mirny, 2019*). In contrast, at sufficiently high processivities ($\lambda$) and linear densities ($1/d$), symmetric two-sided LEFs eliminate unlooped gaps (*Goloborodko et al., 2016b*; *Banigan et al., 2020*). Based on the idea of closing gaps between LEFs, we previously argued that asymmetric two-sided extrusion could fully linearly compact mitotic chromosomes provided that the residence time is sufficiently long; in particular, we require $\lambda_{\text{slow}}/d \gg 1$ (*Banigan and Mirny, 2019*; *Figure 1b*).

In simulations, we controlled asymmetry by varying the relative extrusion speeds of the active subunits, quantified by the dimensionless ratio $v_{\text{slow}}/v_{\text{fast}}$. For all cases of asymmetric two-sided extrusion (i.e. $v_{\text{slow}}>0$) over the simulated range ($10^{-4} \leq v_{\text{slow}}/v_{\text{fast}}<1$), chromosomes can be linearly compacted 1000-fold, provided that $\lambda_{\text{fast}}/d$ (and thus, $\lambda_{\text{slow}}/d$) is sufficiently large (*Figure 1c*). Specifically, fold compaction, $\mathcal{FC}$, grows rapidly for $\lambda_{\text{slow}}/d>1$ (*Figure 1—figure supplement 1a*), as predicted. This occurs because gaps between LEFs are eliminated even if the gap is bordered by two slow LEF subunits (*Figure 1b* and *Figure 1—figure supplement 1b*). As shown in *Figure 1c*, 1000-fold compaction can be achieved for all asymmetries of two-sided LEFs, notably including the asymmetries and $\lambda/d$ in the range of expected physiological values (see below and Materials and methods).

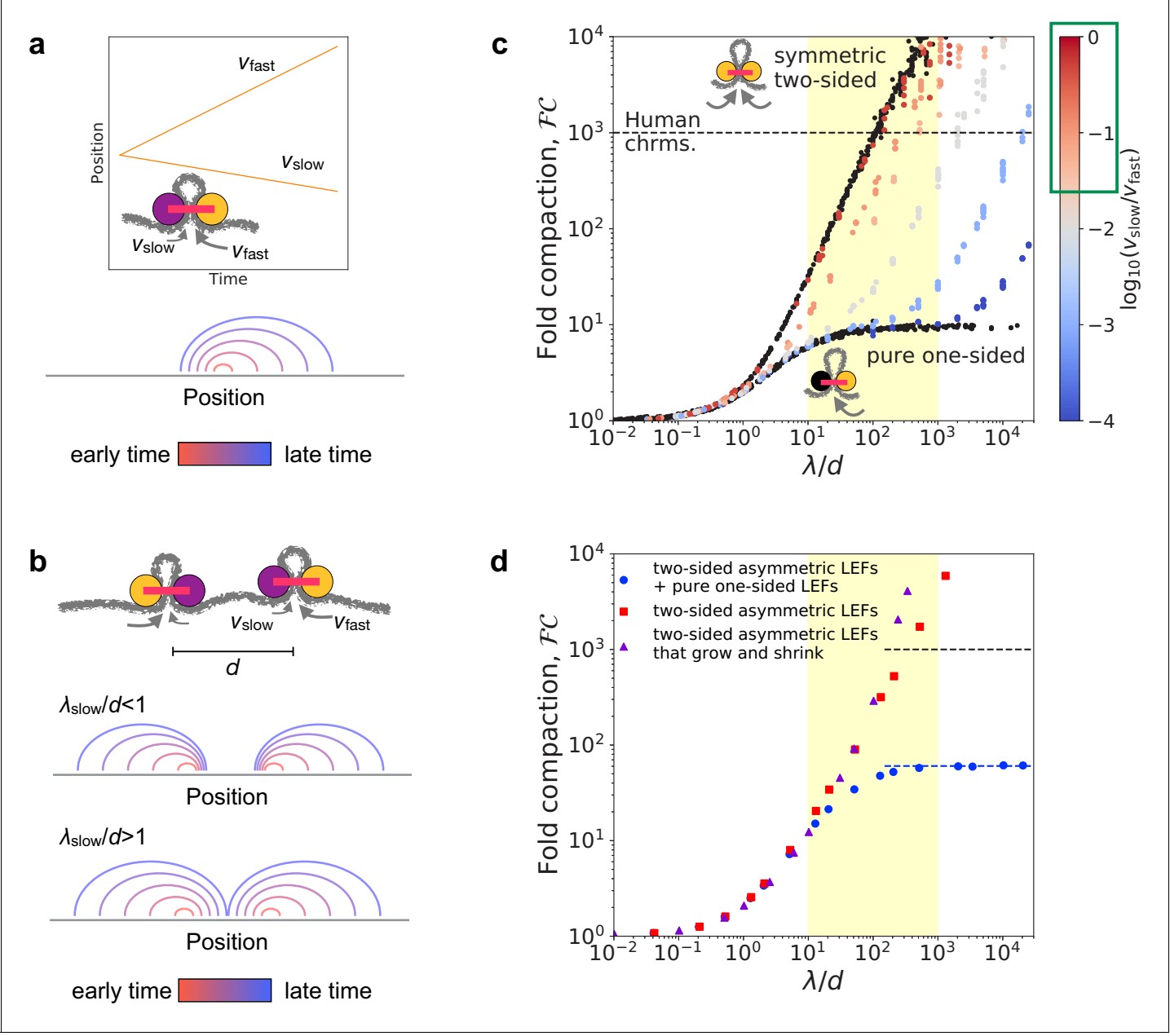

**Figure 1.** Linear compaction by asymmetric two-sided loop extrusion. (**a**) *Top:* Example position versus time trajectory with inset cartoon of an asymmetric two-sided LEF. *Bottom:* Arch diagram for a single asymmetric LEF at several times, with early times in red and late times in blue. (**b**) *Top:* Cartoon of two asymmetric LEFs, oriented so that both slow subunits extrude the chromatin between the two LEFs. *Middle:* Arch diagram showing progressive growth of loops from early times (red) to late times (blue) for LEFs with very slowly extruding 'slow' subunits. *Bottom:* Arch diagram showing gap closure by LEFs with 'slow' subunits that extrude sufficiently rapidly. (**c**) Fold compaction, $\mathcal{FC}$, of asymmetric LEFs, with color indicating the degree of asymmetry, quantified by $\log(v_{slow}/v_{fast})$, from blue (small $v_{slow}$, high asymmetry) to gray to red ($v_{slow} \lesssim v_{fast}$, small asymmetry). For reference, $\mathcal{FC}$ of chromosomes with symmetric two-sided LEFs and one-sided LEFs (i.e. completely asymmetric) are shown in black. Yellow shading indicates expected physiological range for $\lambda/d$ (see text). Green box on the color bar indicates range of asymmetries of 2/3 of metaphase (condensin) loop-extrusion events observed by *Golfier et al., 2020*. Black dashed line indicates 1000-fold linear compaction expected for human mitotic chromosomes. (**d**) Fold compaction, $\mathcal{FC}$, versus $\lambda/d$ for chromosomes with LEF populations with distributions of asymmetries, modeling experimental observations by *Golfier et al., 2020*. Blue circles correspond to 50% asymmetric two-sided LEFs and 50% one-sided LEFs (scenario 1 in the text). Red squares correspond to 100% asymmetric two-sided LEFs with a distribution of asymmetries (scenario 2). Purple triangles correspond to asymmetric two-sided LEFs that can both grow and shrink loops (scenario 3). Blue dashed line indicates theoretical prediction of $\mathcal{FC}_{max} = 60$ for a 50/50 mix of one- and two-sided LEFs (*Banigan and Mirny, 2019*). Yellow shading indicates expected physiological range for $\lambda/d$. Standard error for all displayed data points is <3% of the mean value.

*Figure 1 continued on next page*

*Figure 1 continued*

The online version of this article includes the following figure supplement(s) for figure 1:

**Figure supplement 1.** Compaction and elimination of gaps by asymmetric LEFs.

We next determined whether 1000-fold linear compaction can be achieved with the asymmetries ($v_{slow}/v_{fast}$) observed in single-molecule experiments (*Golfier et al., 2020*) and previously estimated values of $\lambda/d$ for condensin (*Goloborodko et al., 2016b*; *Banigan et al., 2020*). We performed three sets of simulations with asymmetric two-sided LEFs (see Materials and methods): (1) simulations with a 50/50 mixture of asymmetric two-sided LEFs and one-sided LEFs, which has the same distribution of 'symmetry scores' ($S = (v_{fast} - v_{slow})/(v_{fast} + v_{slow})$) as measured for *Xenopus* condensins by *Golfier et al., 2020*, (2) simulations with only asymmetric two-sided LEFs that also reproduce the symmetry score distributions measured by *Golfier et al., 2020*, and (3) simulations with asymmetric two-sided LEFs that can both grow and shrink loops according to our reanalysis of the experiments of *Golfier et al., 2020*.

These three scenarios lead to two qualitatively different outcomes. The mixture of asymmetric two-sided LEFs and one-sided LEFs (scenario 1) can achieve only ~60-fold linear compaction (blue circles in *Figure 1d*). Because half of the LEFs are one-sided in this scenario, a relatively large number of unlooped gaps remain in steady state, which limits linear compaction. In contrast, simulations with different distributions of asymmetric two-sided extrusion (scenarios 2 and 3) do not have this limitation, which results in >1000-fold linear compaction for plausible values of $\lambda/d$ (<1000) (red squares and purple triangles in *Figure 1d*). Although a significant fraction of LEFs are highly asymmetric ($\geq 20\%$ of LEFs with $v_{slow} < 0.1 v_{fast}$), they are typically able to close gaps within their residence times. Thus, we conclude that even highly asymmetric two-sided LEFs can close gaps and compact chromosomes, while a modest amount of strictly one-sided LEFs significantly inhibits compaction.

## Compaction by mixtures of one- and two-sided LEFs depends on their relative dynamic properties

### Model for mixtures of one- and two-sided LEFs with different dynamics

Previous modeling predicts that a large majority (>84%) of LEFs must perform two-sided extrusion in order to sufficiently compact a mitotic chromosome (*Banigan and Mirny, 2019*), but experiments only observe symmetric two-sided extrusion by 20–80% of condensins (*Kong et al., 2020*; *Golfier et al., 2020*). However, previous analyses of mixtures of one- and two-sided LEFs made a potentially important simplifying assumption; they considered only mixtures in which every active subunit translocates along the chromatin fiber at the same speed and every LEF has the same mean residence time (*Banigan and Mirny, 2019*; *Banigan et al., 2020*). In contrast, experimental measurements indicate that the condensins I and II, both of which compact mitotic chromosomes (*Ono et al., 2003*; *Hirota et al., 2004*; *Ono et al., 2004*; *Green et al., 2012*; *Ono et al., 2017*; *Gibcus et al., 2018*; *Hirano, 2016*; *Kalitsis et al., 2017*; *Takahashi and Hirota, 2019*), have different mean residence times (*Gerlich et al., 2006a*; *Walther et al., 2018*) and extrusion speeds (*Kong et al., 2020*).

In FRAP experiments, condensin I and condensin II have markedly different residence times on mammalian mitotic chromosomes. Condensin I typically remains bound to chromosomes for 2–3 min. Condensin II, in contrast, exhibits two types of turnover kinetics; 15–40% of condensins have a mean residence time of 5–8 min, while the remaining complexes are stably bound for longer durations (*Gerlich et al., 2006a*; *Walther et al., 2018*). Furthermore, a recent estimate based on Hi-C analysis and computational modeling suggests a 2-hr residence time for condensin II (*Gibcus et al., 2018*).

Condensins I and II also have different extrusion velocities in vitro. Recent single-molecule experiments (*Kong et al., 2020*), observed that loops extruded by condensin I grow at approximately twice the speed of those extruded by condensin II. Intriguingly, different extrusion speeds are also observed for loop-extruding *cohesins*, depending on whether they perform one-sided or two-sided extrusion (*Golfier et al., 2020*). Together, these results demonstrate that the dynamics of SMC complexes may depend on their associated proteins (as with condensin I versus condensin II) or their mode of extrusion (as with one-sided versus two-sided extrusion).

To determine whether experimentally observed fractions of two-sided condensins can achieve 1000-fold linear compaction, we developed simulation and theory models for mixtures of LEFs with different mean velocities and/or residence times. We primarily consider mixtures of one-sided and two-sided LEFs, where each population has a distinct residence time and extrusion velocity. Further details are provided in the Materials and methods section. *Figure 2a* shows schematic drawings of the LEFs and an arch diagram for an example system.

## Theoretical analysis of mixtures of one- and two-sided LEFs with different dynamics

We developed a theoretical model to quantitatively predict the degree of compaction expected with mixtures of one- and two-sided LEFs with different residence times and extrusion speeds (denoted by subscripts 1 and 2, respectively; see Appendix 1 for the full theory). The theoretical analysis predicts that the ratio of the extrusion speeds, $v_2/v_1$, does not affect the maximum fold linear compaction, $\mathcal{FC}_{\max}$, because the speeds do not affect gap formation (for simulations, see Appendix 2). However, the theory predicts differences between mixtures with very long-lived two-sided LEFs ($\tau_2 \gg \tau_1$) and mixtures with very short-lived two-sided LEFs ($\tau_2 \ll \tau_1$).

In the scenario with long-lived two-sided LEFs ($\tau_2 > \tau_1$), the short-lived one-sided LEFs act as transient barriers to extrusion by the two-sided LEFs (*Figure 2b*, left panel). After a barrier unbinds, the two-sided LEF can extrude beyond that barrier, potentially closing an unlooped gap between one-sided LEFs. The presence of transient barriers reduces the effective speed of the long-lived two-sided LEFs, in turn reducing the effective processivity of the those LEFs to $\lambda_2^{\mathrm{eff}}$. We then numerically compute the fraction of the fiber that we expect to be compacted by a system with only two-sided LEFs at processivity-to-density ratio $\lambda_2^{\mathrm{eff}}/d_2$ (where $d_2$ is the mean separation between two-sided LEFs). The remaining fraction that is not compacted by the two-sided LEFs is assumed to be ~90% compacted by the one-sided LEFs (i.e. the short-lived one-sided LEFs compact the remaining fiber ~10-fold [*Banigan and Mirny, 2019*]). The theory predicts that mixtures with long-lived two-sided LEFs ($\tau_2 > \tau_1$) compact more effectively than mixtures with a single mean residence time ($\tau_2 = \tau_1$).

In the scenario with short-lived two-sided LEFs ($\tau_2 < \tau_1$), long-lived one-sided LEFs act as permanent barriers to extrusion by two-sided LEFs (*Figure 2b*, right panel). In this case, the processivity of the two-sided LEFs is effectively limited to the mean separation, $d_1$, between one-sided LEFs. Following the previous calculation (see Appendix 1), we find that compaction by mixtures with short-lived two-sided LEFs is lower than in mixtures with $\tau_2 = \tau_1$.

These theoretical limits, along with theory developed previously for populations of LEFs with a single residence time ($\tau_2 = \tau_1$) (*Banigan and Mirny, 2019*), establish predictions for the simulations described below; theoretical results are shown in *Figure 2d*.

## Long-lived two-sided LEFs enhance compaction by mixtures of LEFs

We next used simulations to test the prediction that increasing the residence time of the two-sided LEFs relative to that of the one-sided LEFs could increase compaction in mixtures of LEFs. We hypothesized that long-lived two-sided LEFs might be able to further compact chromatin by two mechanisms. First, increasing the residence time increases the effective processivity, and thus potentially, the loop size. Second, increasing the residence time additionally allows two-sided LEFs to remain on the chromatin fiber longer than the one-sided LEFs; consequently, each transient gap between one-sided LEFs may eventually be extruded into a loop by a two-sided LEF.

In both qualitative and quantitative agreement with the theory, the maximum fold linear compaction, $\mathcal{FC}_{\max}$, in simulations increases with increasing two-sided LEF residence times (*Figure 2c–e*, *Figure 2—figure supplement 2a*, and *Figure 2—figure supplement 3*). As predicted, unlooped gaps that are formed by pairs of neighboring one-sided LEFs are short-lived, and thus, they are less common for larger $\tau_2/\tau_1$ (*Figure 2—figure supplement 2b*). Therefore, mixtures of LEFs can achieve >1000-fold compaction with long-lived two-sided LEFs because almost all of the chromatin fiber can be extruded into loops by the two-sided LEFs.

The compaction abilities of mixtures of one- and two-sided LEFs with different residence times is summarized by the phase diagram in *Figure 2e*. The maximum fold compaction, $\mathcal{FC}_{\max}$, depends on both the composition, $\phi_1$, and the relative residence times, $\tau_2/\tau_1$. With very long-lived two-sided LEFs ($\tau_2/\tau_1 > 10$), two-sided LEFs can extrude most of the fiber because gaps formed by the one-

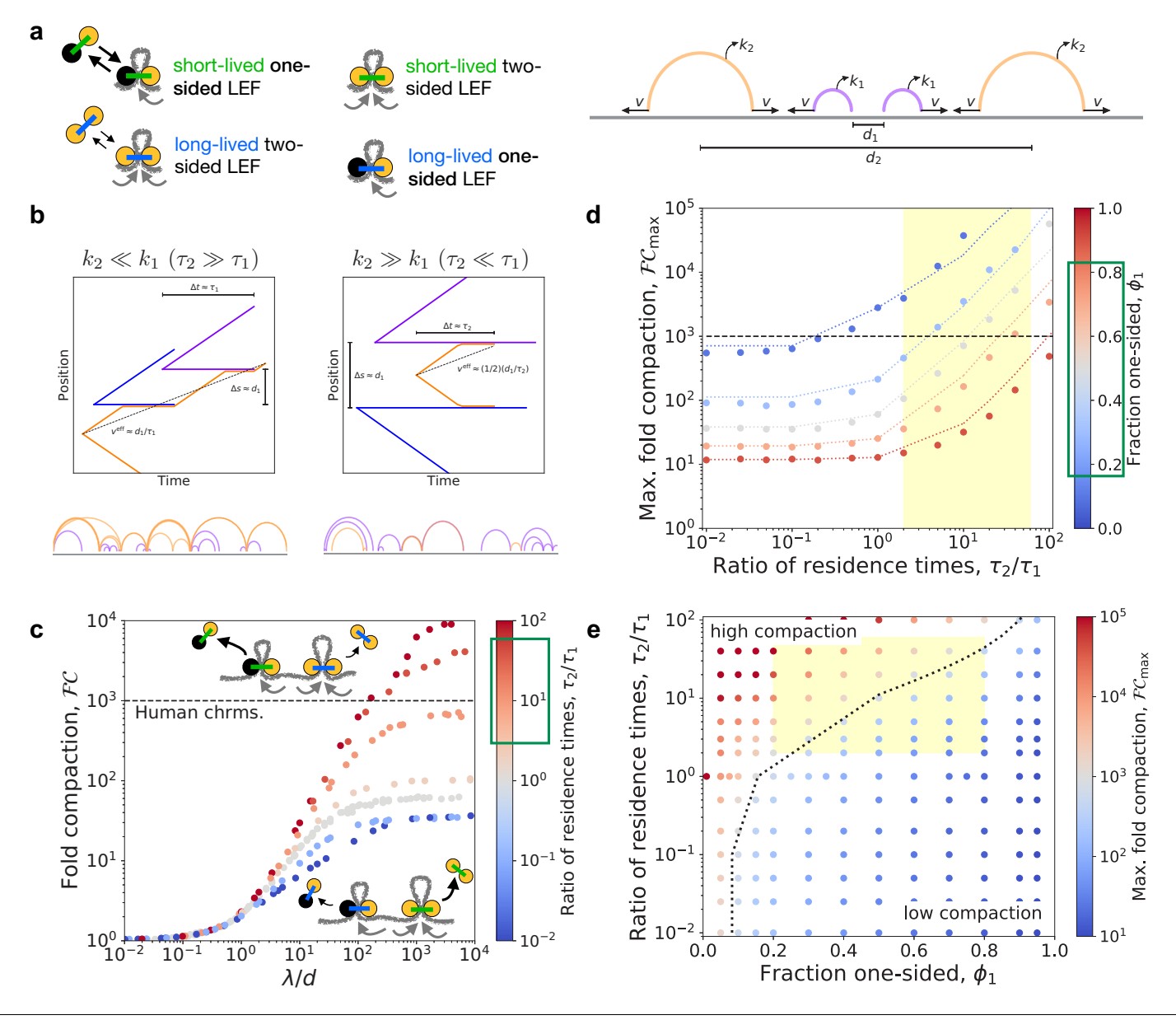

**Figure 2.** Linear compaction by mixtures of one- and two-sided LEFs with different residence times. (**a**) Arch diagram schematically showing two-sided LEFs (orange) and one-sided LEFs (purple) with different unbinding rates (and residence times), $k_2 = 1/\tau_2$ and $k_1 = 1/\tau_1$, respectively. (**b**) *Top:* Example trajectories for mixtures of one- and two-sided LEFs illustrating the theoretical limits of very long-lived two-sided LEFs (left) and very short-lived two-sided LEFs (right). In both scenarios, a two-sided LEF (orange) extrudes at speed $v_2$ until encountering a one-sided LEF (blue or purple), which acts as a barrier. Encounters with barriers reduce the effective mean speed, $v^{eff}$, indicated by black dashed lines. *Bottom:* Arch diagrams from simulation snapshots for the two scenarios. Two-sided LEFs are orange and one-sided LEFs are purple. (**c**) Fold-compaction, $\mathcal{FC}$, as a function of the ratio, $\lambda/d$, for 50/50 mixes of one- and two-sided LEFs in which each species has a different residence time. Color from blue to gray to red indicates increasing ratio, $\tau_2/\tau_1 = k_1/k_2$, of residence times. Green box on the color bar indicates estimated range of residence time ratios for condensin II as compared to condensin I (*Gerlich et al., 2006a*; *Gibcus et al., 2018*; *Walther et al., 2018*). Gray points are data for $\tau_2 = \tau_1$. Black dashed line indicates 1000-fold compaction. (**d**) Maximum fold compaction, $\mathcal{FC}_{max}$, as a function of the ratio of residence times, $\tau_2/\tau_1$. Colors from blue to gray to red indicate increasing fractions, $\phi_1$, of one-sided LEFs. Dotted lines indicate theory, with results for $\tau_2 \ll \tau_1$ shown for $\tau_2/\tau_1 < 0.1$, results for $\tau_2 \gg \tau_1$ shown for $\tau_2/\tau_1 > 10$, and the mean-field ($\tau_2/\tau_1 = 1$) result shown with straight lines interpolating to other theoretical results. Yellow region indicates estimated range of residence time ratios for condensin II as compared to condensin I. Green box indicates approximate range of $\phi_1$ observed in experiments with metazoan condensins (*Golfier et al., 2020*; *Kong et al., 2020*). (**e**) Phase diagram colored by fold linear compaction, $\mathcal{FC}$, for various fractions, $\phi_1$, of one-sided LEFs and ratios, $\tau_2/\tau_1$ of residence times (red: high compaction, blue: low compaction, gray: $\mathcal{FC} \approx 1000$). Yellow region indicates range of

*Figure 2 continued on next page*

*Figure 2 continued*

residence time ratios for condensin II as compared to condensin I and experimentally observed fractions of one-sided extruders. Black dotted line is the theoretically predicted boundary ($\mathcal{FC} = 1000$) between high and low compaction regimes. Standard error for all displayed data points is <3%.

The online version of this article includes the following figure supplement(s) for figure 2:

**Figure supplement 1.** Compaction by mixtures of one- and two-sided LEFs with different extrusion velocities.
**Figure supplement 2.** Compaction by mixtures of one- and two-sided LEFs with different residence times.
**Figure supplement 3.** Compaction by mixtures of one- and two-sided LEFs with different extrusion speeds and residence times.
**Figure supplement 4.** Maximum fold compaction by mixtures one-sided LEFs with different residence times.

sided LEFs are relatively transient and infrequent. In this case, 1000-fold compaction can be achieved even with fairly large fractions, $\phi_1$, of one-sided LEFs; for example, with $\tau_2/\tau_1 \approx 40$, up to ~70% of LEFs may be one-sided. In contrast, with short-lived two-sided LEFs ($\tau_2/\tau_1 < 1$), a large fraction, $\phi_2$, of two-sided LEFs is required to achieve 1000-fold compaction because the two-sided LEFs are frequently impeded by the long-lived one-sided LEFs; therefore, many two-sided LEFs are needed to fully extrude the gaps between one-sided LEFs (e.g. $\phi_1 = 0.7$ now results in $\mathcal{FC} < 25$, and $\mathcal{FC} = 1000$ requires $\phi_1 < 0.16$). Between these limits ($1 < \tau_2/\tau_1 < 10$), systems with moderate fractions of one-sided LEFs (e.g. $0.16 < \phi_1 < 0.5$) can achieve 1000-fold linear compaction. These results establish that mixtures of LEFs can fully compact chromosomes provided that high fractions of one-sided LEFs are adequately offset by long residence times for two-sided LEFs.

## LEFs with the dynamics of condensins I and II can compact chromosomes

We next considered our results for relative residence times ($\tau_2/\tau_1$) and fractions of one-sided extruders ($\phi_1$) measured by and estimated from experiments. FRAP experiments and Hi-C experiments and modeling suggest a range of $2 \leq \tau_2/\tau_1 \leq 60$ for the ratio of condensin II to condensin I residence times (*Gerlich et al., 2006a*; *Gibcus et al., 2018*; *Walther et al., 2018*) (yellow region in *Figure 2d and e*); single-molecule experiments with metazoan condensins suggest a fraction of one-sided condensins in the range $0.2 \leq \phi_1 \leq 0.8$ green box in *Figure 2d* and yellow region in *Figure 2e* (*Golfier et al., 2020*; *Kong et al., 2020*). These ranges suggest that 1000-fold compaction can be achieved within plausible physiological ranges of $\tau_2/\tau_1$ and $\phi_1$.

However, the actual experimental situation is considerably more complicated. In vitro single-molecule experiments with human condensins I and II show that condensins of both types may be either one-sided or two-sided (*Kong et al., 2020*); thus, some one-sided condensins may be long-lived and some two-sided condensins may be short-lived. To address this scenario, we simulated mixtures of LEFs with the properties of condensins I and II (*Figure 3a*, blue box). LEFs representing condensin I were short-lived, with residence time $\tau_I$, and LEFs representing condensin II were longer-lived, with residence time $\tau_{II} = 3\tau_I$. Furthermore, each population of condensin is itself a mixture of one-sided and two-sided LEFs; respectively, 80% and 50% of condensin I and condensin II LEFs were two-sided, as in single-molecule experiments (*Kong et al., 2020*). To match experimental measurements in *Xenopus* and HeLa cells (*Ono et al., 2003*; *Shintomi and Hirano, 2011*; *Walther et al., 2018*), we assume 80% of LEFs are condensin I (as a result, 74% of all LEFs are two-sided).

With this base model for mixtures of condensins I and II, we simulated chromosome compaction. We first found that the mixtures of condensins described above can generate only ~260-fold linear compaction, less than the 1000-fold compaction required for human mitotic chromosomes (blue circles in *Figure 3b*).

We then noted that in vivo, condensin II has two subpopulations with different residence times; 15–40% of all condensin II dynamically exchange with a mean residence time of 5–8 min, while the remaining 60–85% of condensin II complexes are stably bound for a much longer residence time (*Gerlich et al., 2006a*; *Walther et al., 2018*). Therefore, we simulated a modified condensin model in which 50% of the condensin II LEFs are more stably bound ('extra-stable') with a mean residence time of $\tau_{II,*} = 20\tau_{II} = 60\tau_I$ (the underestimate of 50% stably bound is for simplicity; see scenario below) (*Figure 3a*, red box). In these simulations, condensin I and II mixtures can generate up to ~540-fold linear compaction, still short of our expectation for mitotic chromosomes (red squares in *Figure 3b*).

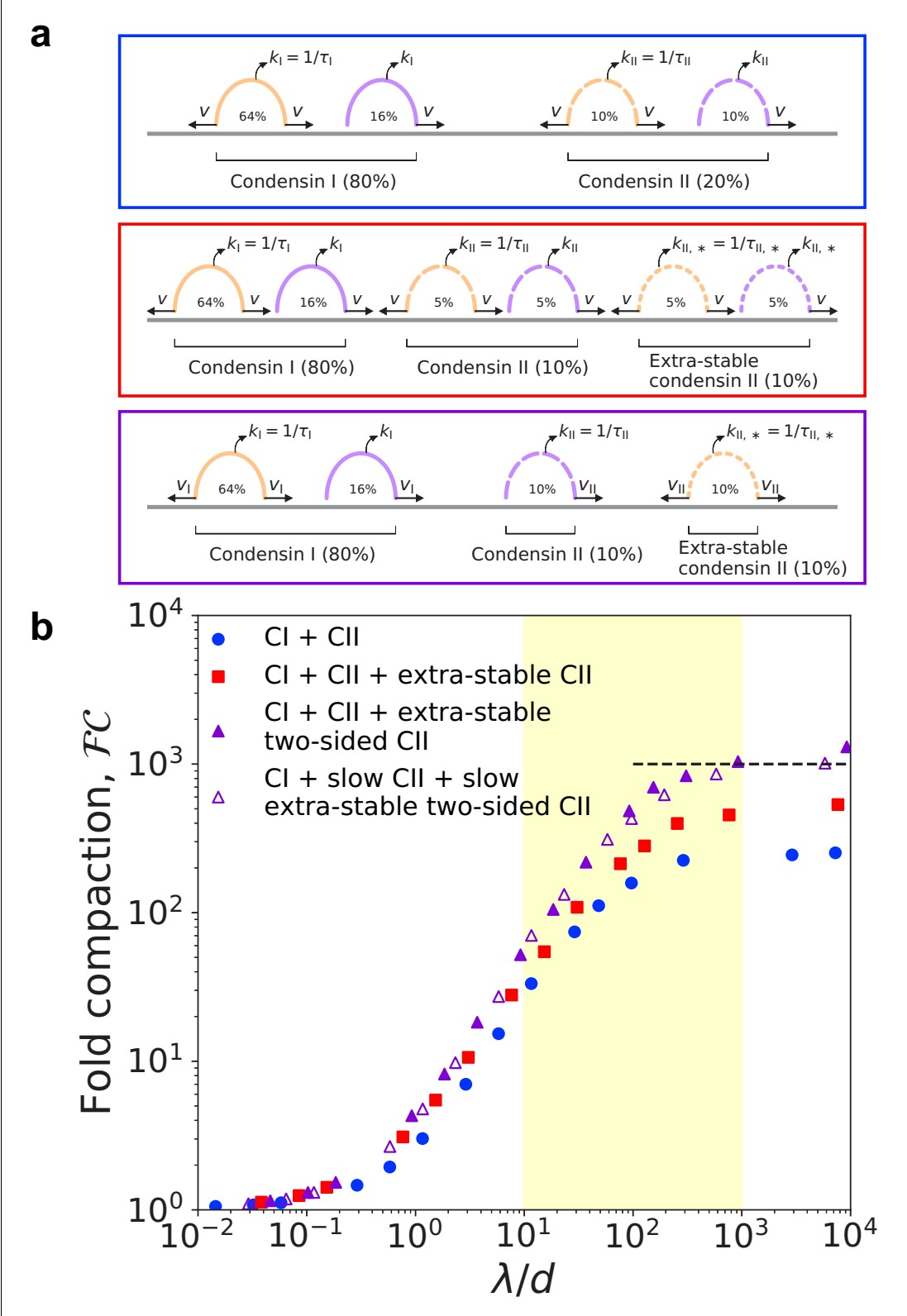

**Figure 3.** Simulations of LEFs with properties of condensins I and II. (**a**) Arch diagrams schematically illustrating three different scenarios for extrusion by mixtures of condensins I and II. Condensins I and II may be one- or two-sided in all scenarios (purple or orange, respectively). The ratio of condensin I to condensin II is 4:1. *Top (blue box):* Schematic cartoon of a mixture of condensins I and II in which condensin II has residence time $\tau_{II} = 3\tau_I$. *Middle (red box):* Cartoon of scenario with two populations of condensin II. The 'extra-stable' population has residence time $\tau_{II,*} = 20\tau_{II} = 60\tau_I$. In this

*Figure 3 continued on next page*

*Figure 3 continued*

scenario, both types of condensin II may be one-sided or two-sided. *Bottom (purple box):* Scenario in which extra-stable condensin II complexes are two-sided, while the more dynamic condensin II subpopulation is one-sided. Colors of boxes indicate color of corresponding data points in panel (**b**).
(**b**) Fold linear compaction, $\mathcal{FC}$, versus $\lambda/d$ for the scenarios described above (blue circles, red squares, and purple triangles, respectively). Open triangles denote the third scenario, but with condensin II complexes extruding with speed $v_{\mathrm{II}} = 0.5v_{\mathrm{I}}$. Black dashed line denotes 1000-fold compaction. Yellow shading indicates expected physiological range of $\lambda/d$. Standard error for all displayed data points is <3%.

The online version of this article includes the following figure supplement(s) for figure 3:

**Figure supplement 1.** Arch diagrams for simulations with LEFs with the properties of condensins I and II.

**Figure supplement 2.** Condensins I and II simulations with different ratios of condensin I to condensin II.

---

To further increase the degree of compaction, we assumed that stably bound condensin II performs two-sided extrusion, while dynamic condensin II performed one-sided extrusion (*Figure 3a*, purple box). In this scenario, mixtures of condensins I and II can generate >1000-fold linear compaction (purple triangles in *Figure 3b*). Extra-stable condensin II LEFs form large loops, while short-lived condensins I LEFs form smaller, nested loops (*Figure 3—figure supplement 1*). These results hold for simulations modeling other cell types with higher or lower levels of condensin I relative to condensin II (*Figure 3—figure supplement 2*), which model mitotic chromosomes in other types of cells (*Ono et al., 2003*; *Ohta et al., 2010*). Altogether, the simulations demonstrate that a coupling between long residence times and two-sided extrusion (and between shorter residence times one-sided extrusion) can enhance the attainable degree of compaction, including in experimentally relevant scenarios.

## Discussion

A key outstanding question for loop-extruding SMC complexes is how predominantly asymmetric extrusion, observed in vitro, can generate the high degree of linear compaction observed for mitotic chromosomes in vivo. We previously argued that effectively two-sided extrusion or a strong targeted loading bias is needed to compact and organize chromosomes (*Banigan and Mirny, 2019*; *Banigan et al., 2020*; *Banigan and Mirny, 2020*). Recent experiments provide evidence that condensins might perform two-sided extrusion (*Golfier et al., 2020*; *Kong et al., 2020*), albeit not precisely in the manner envisioned in previous theoretical arguments (*Banigan and Mirny, 2020*). Our present work establishes how metazoan mitotic chromosomes can be linearly compacted 1000-fold by condensins performing asymmetric two-sided extrusion or by condensins in a predominantly one-sided mixture (*Figure 4*).

First, LEFs performing asymmetric two-sided extrusion as observed in *Xenopus* extracts (*Golfier et al., 2020*) could compact mitotic chromosomes 1000-fold if their 'slow sides' extrude quickly enough to eliminate unlooped gaps (*Figures 1* and *4b*). Second, mixtures of one- and two-sided LEFs in which the two-sided LEFs have relatively long residence times can linearly compact chromosomes 1000-fold, even with large fractions of one-sided LEFs (*Figures 2* and *4c*). Third, in order to achieve 1000-fold compaction with mixtures of condensins I and II, we predict that stable chromatin binding by condensin II complexes (*Gerlich et al., 2006a*; *Walther et al., 2018*) must be coupled to two-sided extrusion (*Figures 3* and *4d*).

Our results for asymmetric LEFs show that LEFs performing asymmetric two-sided extrusion, as in *Xenopus* (*Golfier et al., 2020*), could fully compact mitotic chromosomes (*Figure 1d*). The magnitudes and distribution of asymmetries observed in vitro ($v_{\mathrm{slow}}/v_{\mathrm{fast}}>10^{-2}$) are quantitatively consistent with the condition that gaps between LEFs must be closed within a single residence time ($\lambda_{\mathrm{slow}}/d>1$; *Figure 1b–d* and *Figure 1—figure supplement 1*). Thus, asymmetric two-sided extrusion with one rapidly extruding side ($v_{\mathrm{fast}} \sim 1~\mathrm{kb/s}$) and one slowly extruding side ($10~\mathrm{bp/s}<v_{\mathrm{slow}}<v_{\mathrm{fast}}$) could compact mitotic chromosomes (*Figure 4b*, right).

In mixtures of one- and two-sided LEFs, a longer residence time for two-sided LEFs allows those LEFs to extrude the gaps between one-sided LEFs into loops after the one-sided LEFs unbind (*Figure 2a and b* and right panel of *Figure 4c*). Thus, 1000-fold compaction can be achieved even if the fraction of one-sided LEFs exceeds the $\phi_1 \approx 0.16$ threshold fraction previously predicted for systems with a single mean LEF residence time (*Banigan and Mirny, 2019*; *Figure 2d and e*). In systems with only one-sided LEFs, differences in the mean residence time can enhance linear

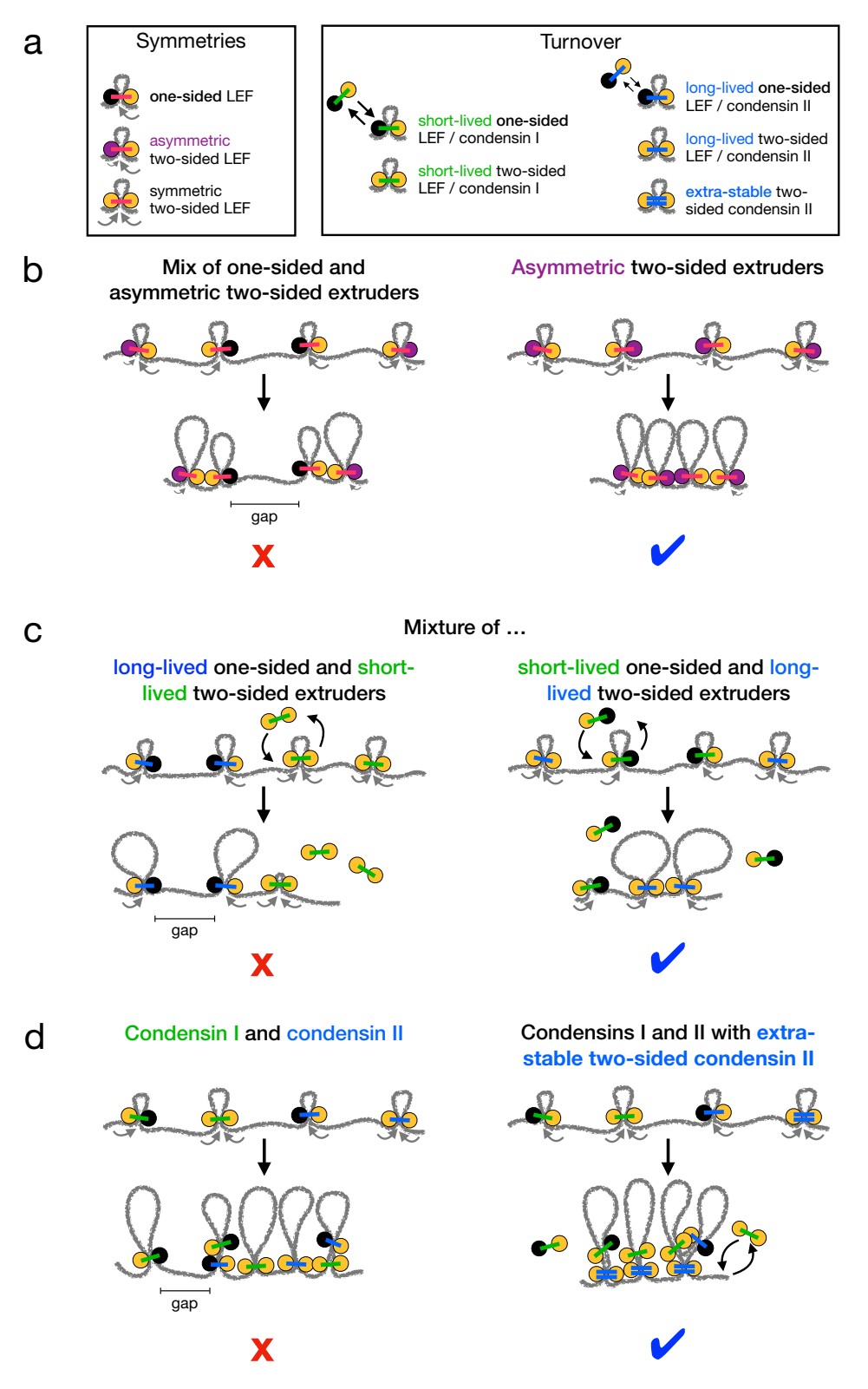

**Figure 4.** Summary of chromosome compaction abilities of LEF mixtures with different asymmetries and dynamics. (**a**) Legend for illustrations of LEFs with different symmetries (left) and/or different dynamics (right). (**b**) Illustrations of systems with asymmetric LEFs. Systems with both one-sided and asymmetric two-sided LEFs do not fully compact mitotic chromosomes due to the unavoidable presence of unlooped gaps (left). However, asymmetric two-sided LEFs can fully compact chromosomes if the processivity of the slow side (purple) is sufficiently large (right). (**c**) Illustrations of one- and two-
*Figure 4 continued on next page*

*Figure 4 continued*

sided LEFs with different dynamics. Mixtures in which one-sided LEFs are long-lived (slowly turn over) and two-sided LEFs are short-lived (rapidly turn over) cannot fully compact mitotic chromosomes (left). Mixtures with long-lived two-sided LEFs and short-lived one-sided LEFs compact chromosomes (right). (d) Illustrations of mixtures of condensins I and II. Mixtures of condensins I and II in which turnover is independent of extrusion symmetry cannot fully compact chromosomes (left). Mixtures in which two-sided condensin II complexes are very long-lived (LEFs with double blue bars) can fully compact chromosomes (right). Note that each LEF represents a molecular complex that performs either one-sided or two-sided extrusion; a two-sided LEF may represent either a single two-sided condensin or a dimer of two one-sided condensins.

compaction, but only to a ~90-fold limit, which is still too small (Appendix 1, Appendix 2, and *Figure 2—figure supplement 4b and c*). Importantly, for mixtures of one- and two-sided LEFs, 1000-fold compaction may be achieved within ranges of residence times expected from experimental observations and computational modeling (*Gerlich et al., 2006a*; *Walther et al., 2018*; *Gibcus et al., 2018*; *Figure 2d and e*).

These results are particularly relevant for condensins I and II, which have different residence times. According to a recent single-molecule experiment, both condensins I and II may perform one-sided and two-sided extrusion (*Kong et al., 2020*). As a result, a difference between the residence times of condensins I and II is insufficient to explain chromosome compaction by itself because some one-sided condensins could be long-lived (*Figure 3a* and left panel of *Figure 4d*). However, if two-sided extrusion by condensin II is coupled to the extremely long residence times observed for a subpopulation of condensin II in vivo (*Gerlich et al., 2006a*; *Walther et al., 2018*), complete mitotic chromosome compaction may be achieved (*Figure 3b*, *Figure 3—figure supplement 2*, and right panel of *Figure 4d*).

These results demonstrate the importance of long residence times together with two-sided extrusion – whether it be symmetric or asymmetric – to robust chromosome compaction. They have several possible implications for the molecular mechanisms of loop extrusion and chromosome organization by SMC complexes.

## Mechanisms for asymmetric two-sided extrusion

It is unclear how a condensin complex could perform asymmetric two-sided extrusion, while still compacting chromosomes in vivo. For instance, diffusive sliding of one side combined with active translocation by the other is unable to form large DNA loops in vitro (*Ganji et al., 2018*), and it is insufficient to consistently eliminate chromatin gaps and achieve 1000-fold compaction in simulations (*Banigan et al., 2020*). Instead, complete compaction requires directed translocation to collect chromatin from both sides of the condensin complex. Directed translocation could be governed by one or more of several mechanisms.

One possibility is that asymmetric two-sided extrusion could occur due to asymmetry that is intrinsic to the complex. The degree of asymmetry of a condensin could be controlled by the species of the kleisin or Hawks (HEAT proteins associated with kleisins) associated with the complex. Consistent with this hypothesis, it has been observed that the yeast kleisin Brn1 and Hawk Ycg1 can act as a 'safety belt' that anchors condensin to DNA (*Kschonsak et al., 2017*). However, anchoring by the safety belt combined with condensin translocation (*Terakawa et al., 2017*) generates pure one-sided loop extrusion (*Ganji et al., 2018*), which is incompatible with the high degree of mitotic chromosome compaction observed in metazoan cells (*Banigan and Mirny, 2019*). Conceivably, a looser safety belt combined with the ability of the complex to perform two-sided extrusion could generate asymmetric two-sided extrusion in higher eukaryotes.

Alternatively, complexes performing asymmetric two-sided extrusion could be dimers of condensins, for which asymmetries could arise by several mechanisms. The extruding complex could contain condensins with two different protein compositions. Each side of the resulting complex might have different extrusion dynamics. However, this possibility is diminished by the strong tendency for particular kleisins and HAWKs to associate together (i.e., form either condensin I or II) (*Ono et al., 2003*) and the different patterns of spatial localization of these proteins (*Ono et al., 2003*; *Shintomi et al., 2017*; *Walther et al., 2018*). A more likely possibility is that the compositions of the condensins within a dimer are identical, but asymmetry arises if the components are differentially regulated by post-translational modifications. This would be consistent with the observation of asymmetric two-sided extrusion by *Xenopus* condensins in their native context (*Golfier et al., 2020*). Yet

another possibility is that dimerization of condensins generates an asymmetric complex. This could arise due to chirality within the joined molecules or through differential conformational changes that are required to form the dimer. These scenarios are not mutually exclusive; several of these mechanisms could act together to generate asymmetric two-sided extrusion.

## Mechanisms for stability of two-sided condensins

What could give rise to the predicted stability of two-sided SMC complexes? One hypothesis is that condensin I performs one-sided extrusion, while condensin II performs two-sided extrusion (*Banigan and Mirny, 2020*). The mean residence time of dynamic condensin II is about three times longer than that of condensin I, and there is an immobile subpopulation of condensin II with a much longer residence time (*Gerlich et al., 2006a*; *Walther et al., 2018*). Thus, two-sided complexes (condensin II in this scenario) would be long-lived compared to one-sided complexes (condensin I). Such a mixture of condensins I and II could generate 1000-fold chromosome compaction (*Figures 2* and *4c*). Further supporting this hypothesis, yeast condensin, which is evolutionarily conserved as condensin I (*Hirano, 2012*), performs one-sided loop extrusion in vitro (*Ganji et al., 2018*). Thus, it is appealing to think that condensin II is a two-sided counterpart to one-sided condensin I in metazoan cells.

However, single-molecule experiments with condensins I and II suggest otherwise, at least for human cells (*Kong et al., 2020*). It has been observed that human condensins I and II can both perform either one-sided or two-sided extrusion in vitro. Moreover, two-sided extrusion events are observed more frequently for condensin I as compared to condensin II. Interestingly, photobleaching experiments show that two-sided loop-extruding condensin I complexes are dimers of condensin I molecules (*Kong et al., 2020*). This observation suggests that perhaps individual condensins are one-sided extruders, but they can dimerize to form two-sided loop-extruding complexes.

Taking the in vitro experiments (*Kong et al., 2020*) together with our simulation results (*Figure 2e* and *Figure 3b*), we propose that condensin II complexes dimerize to form two-sided loop-extruding complexes with very long residence times. Dimerization could facilitate two-sided extrusion by combining two one-sided complexes, while also stabilizing binding of condensin II to DNA by increasing (or otherwise modifying) the condensin-DNA binding surface. This combination of effects would naturally generate a tight coupling between two-sided extrusion and long residence times. Thus, dimerization could simultaneously allow two-sided extrusion and increase the residence time. This combination would generate the necessary conditions for chromosome compaction by mixtures of one- and two-sided condensins.

## Spatial organization by condensins I and II

Metazoan chromosomes are composed of large (~400 kb) chromatin loops with condensin II at their bases, with smaller (~80 kb) loops mediated by condensin I nested within (*Gibcus et al., 2018*; *Walther et al., 2018*). Consequently, condensin II is tightly localized to the central axis of the rod-like chromosome, while condensin I is localized along the axis in a broader pattern (*Ono et al., 2003*; *Shintomi et al., 2017*; *Walther et al., 2018*).

This hierarchy of loop nesting naturally emerges from the loop extrusion model for mixtures of LEFs with two different residence times. LEFs with longer residence times, $\tau$, have a larger processivities, $\lambda = v\tau$, and thus tend to form larger loops. In 3D, the bases of these loops localize along a central axis (*Goloborodko et al., 2016a*; *Gibcus et al., 2018*). This hierarchy emerges in mixtures of LEFs with the properties of condensins I and II (*Figure 3—figure supplement 1*), mixtures of one- and two-sided LEFs (*Figure 2b*, bottom left), mixtures of two populations of one-sided LEFs (*Figure 2—figure supplement 4e*), and mixtures of two populations of two-sided LEFs (*Gibcus et al., 2018*). While not all combinations of LEF symmetries and dynamics can fully compact mitotic chromosomes (two-sided extrusion is required; *Figure 4*), differences in residence times can generate the nesting structure required for the patterns of spatial localization of condensins I and II observed in vivo. Consistent with this idea, the residence time of condensin II in vivo is longer than that of condensin I (*Gerlich et al., 2006a*; *Walther et al., 2018*).

Furthermore, our results suggest specific properties of condensins that could regulate mitotic chromosome morphology. Previously, it was shown that mitotic chromosome morphology can be controlled by the relative ratio of condensin I to condensin II (*Shintomi and Hirano, 2011*).

Generally, condensin II is responsible for lengthwise 'axial' compaction along the central axis of a chromatid, while condensin I drives 'lateral' compaction, reducing the width of a chromatid (*Ono et al., 2003*; *Shintomi and Hirano, 2011*; *Green et al., 2012*; *Bakhrebah et al., 2015*; *Shintomi et al., 2017*; *Hirano, 2016*; *Kalitsis et al., 2017*). We find that morphology and the degree of linear compaction could alternatively be controlled by the relative residence times of condensins I and II and the fraction of condensins that perform one-sided (instead of two-sided extrusion). For example, a perturbation (such as a post-translational modification) that increases the residence time of condensin II could lead to greater lengthwise compaction due to the larger loops that would be formed by condensin II. Similarly, perturbations that promote two-sided extrusion, especially by condensin II, could also increase the degree of lengthwise compaction. Moreover, molecular perturbations affecting the coupling between symmetry and residence time could have a significant effect on compaction and morphology. These types of biomolecular perturbations would enable cells to regulate chromosome compaction without requiring global changes to levels of condensin expression.

## Kinetics of compaction by condensins I and II

In the loop extrusion model, LEFs can linearly compact chromosomes by extruding chromatin into loops within approximately one LEF residence time. Chromosome morphology, as quantified by loop sizes, equilibrates over longer timescales of approximately 5–10 residence times (*Goloborodko et al., 2016a*; *Goloborodko et al., 2016b*; *Banigan et al., 2020*). In mixtures of LEFs with different dynamics, there are multiple timescales that could govern compaction kinetics. Linear compaction into a series of loops may still occur within ~1–2 residence times of the short-lived LEFs, but equilibration of large loops formed by the long-lived LEFs will occur over several residence times of the long-lived LEFs.

With the experimentally measured residence times of condensins I and II (*Gerlich et al., 2006a*; *Walther et al., 2018*), loop extrusion could compact chromosomes into loop arrays within a few minutes during prophase (*Gibcus et al., 2018*). However, loops should continue to merge and grow as mitosis progresses since the residence time for stably bound condensin II is comparable to the duration of mitosis (*Gerlich et al., 2006a*; *Walther et al., 2018*). Consequently, chromosomes should axially shorten and undergo 3D compaction as mitosis progresses, as observed in vivo (*Nagasaka et al., 2016*; *Gibcus et al., 2018*). Simultaneously, chromosomes should be further compacted by progressive loading of condensins, which increases throughout mitosis (*Walther et al., 2018*) and hyper-compacts chromosomes when mitosis is stalled (*Sun et al., 2018*). Thus, even with mixtures of condensins with long residence and equilibration times, 1000-fold linear compaction is achievable during mitosis.

## Expectations for extrusion in other scenarios

The principles for mitotic chromosome compaction developed here are generalizable to other chromosome organization scenarios. In particular, we previously showed that the physical principles underlying mitotic chromosome formation by loop extrusion are relevant for topologically associated domain (TAD) formation in interphase (*Banigan et al., 2020*), where extrusion by the SMC complex cohesin can be paused by barriers such as CTCF (*Sanborn et al., 2015*; *de Wit et al., 2015*; *Fudenberg et al., 2016*; *Busslinger et al., 2017*; *Nora et al., 2017*; *Wutz et al., 2017*). Thus, we expect that asymmetric two-sided LEFs and mixtures of LEFs might be able to form TADs. There are similar requirements for juxtaposition of bacterial chromosome arms by bacterial SMC complexes, but there also are additional constraints due to the specific loading site near the origin of replication (*Banigan et al., 2020*). Accordingly, asymmetric two-sided LEFs and mixtures of LEFs with different residence times might be able to form TADs but not be able to juxtapose chromosome arms.

### Topologically associated domains

We previously showed that much like mitotic chromosome compaction, the formation of major features of interphase chromosomes, such as TADs, 'dots', and 'stripes' requires avoiding unlooped gaps, either between LEFs or between LEFs and TAD boundaries. One-sided extrusion can form TADs and stripes by enhancing local chromatin contacts (*Banigan et al., 2020*), as observed in Hi-C experiments (*Dixon et al., 2012*; *Nora et al., 2012*; *Sexton et al., 2012*; *Rao et al., 2014*;

*Fudenberg et al., 2016*; *Vian et al., 2018*; *Barrington et al., 2019*). However, dots (*Rao et al., 2014*; *Krietenstein et al., 2020*) can only be generated by 'effectively two-sided' loop extrusion because such extrusion can reliably bring together TAD boundaries (e.g. convergently oriented CTCF binding sites [*Rao et al., 2014*; *Guo et al., 2015*; *Sanborn et al., 2015*; *de Wit et al., 2015*; *Vietri Rudan et al., 2015*]). In TADs, asymmetric two-sided LEFs should be able to eliminate unlooped gaps if the slow side of each LEF is fast enough. Specifically, TAD boundaries could be brought together if the processivity, $\lambda_{\text{slow}}$, of the slow side is larger than either the mean distance between LEFs ($d$) or the TAD size ($L_{\text{TAD}}$). We expect $\lambda = \lambda_{\text{fast}} + \lambda_{\text{slow}} \sim 100 - 1000 \text{ kb}$, $d \sim 100 - 200 \text{ kb}$, and $L_{\text{TAD}} \sim 100 - 1000 \text{ kb}$, based on previous simulations (*Fudenberg et al., 2016*; *Banigan et al., 2020*), measurements of cohesin's properties, (*Davidson et al., 2019*; *Kim et al., 2019*; *Golfier et al., 2020*; *Gerlich et al., 2006b*; *Kueng et al., 2006*; *Tedeschi et al., 2013*; *Hansen et al., 2017*; *Wutz et al., 2017*; *Cattoglio et al., 2019*; *Holzmann et al., 2019*), and Hi-C maps (*Dixon et al., 2012*; *Nora et al., 2012*; *Sexton et al., 2012*; *Rao et al., 2014*). These values suggest that asymmetric two-sided loop extrusion by cohesin could generate TADs, dots, and stripes for moderate asymmetries ($v_{\text{slow}}/v_{\text{fast}} > 0.1$). Consistently, DNA loop extrusion by cohesin in vitro is largely symmetric (*Davidson et al., 2019*; *Kim et al., 2019*; *Golfier et al., 2020*).

Mixtures of one- and two-sided cohesins with different residence times should also be able to form TADs. We previously showed that mixtures with relatively high fractions of one-sided LEFs ($\phi_1 \sim 0.5$) could form TADs. Long residence times for the two-sided extruders could enhance the ability of mixtures to bring together TAD boundaries; as in simulations of mitotic chromosomes, short-lived one-sided LEFs would merely form transient barriers to two-sided extrusion. Such a scenario, however, remains largely hypothetical since extrusion by cohesin is mainly two-sided (*Davidson et al., 2019*; *Kim et al., 2019*; *Golfier et al., 2020*).

## Juxtaposition of bacterial chromosome arms

Juxtaposition of bacterial chromosome arms can be achieved in a more limited set of loop extrusion scenarios. Loading of SMC complexes near the origin of replication breaks the translational symmetry of the system; thus, LEFs must extrude loops symmetrically (or nearly so) (*Banigan et al., 2020*).

Consequently, we do not expect that asymmetric two-sided LEFs could produce the patterns observed in Hi-C maps of *Bacillus subtilis* and *Caulobacter crescentus* chromosomes (*Umbarger et al., 2011*; *Le et al., 2013*; *Marbouty et al., 2015*). There, symmetric two-sided extrusion produces a secondary diagonal that is perpendicular to the main (self-contact) diagonal (*Umbarger et al., 2011*; *Le et al., 2013*; *Marbouty et al., 2015*; *Wang et al., 2017*; *Miermans and Broedersz, 2018*; *Banigan et al., 2020*). For bacterial chromosomes, each asymmetric LEF would juxtapose sites separated by different genomic distances, $s_1 \neq s_2$, from the loading site. Asymmetric juxtaposition by many LEFs would thus generate two secondary diagonals in Hi-C maps, where both diagonals would not be perpendicular to the main diagonal.

For similar reasons, mixtures of one- and two-sided LEFs with different residence times generally will not juxtapose bacterial chromosome arms. Each one-sided LEF brings one chromosomal arm into contact with the loading site and interferes with juxtaposition by LEFs that bind subsequently (*Banigan et al., 2020*). Therefore, any substantial level of one-sided extrusion would disrupt chromosomal arm juxtaposition. Such interference in mixtures of one- and two-sided LEFs might be partially mitigated in scenarios in which LEFs may traverse each other (e.g. form Z-loops). However, this possibility requires further detailed investigation since loop extrusion with LEF traversal may lead to a variety of complicated bacterial chromosome structures in simulations and in vivo (*Brandão et al., 2020*) and is subject to ongoing investigation (*Anchimiuk et al., 2020*).

## Conclusion

Two ingredients are essential for mitotic chromosome compaction by condensins: sufficiently long residence times and some amount of either symmetric or asymmetric (effectively) two-sided extrusion. Strikingly, the presence of even a small (~20%) fraction of such condensins in a mixture with other, purely one-sided condensins could be sufficient to achieve compaction under physiological conditions. It remains to be determined what mechanisms are responsible for different types of extrusion dynamics and what factors might facilitate a transition from one-sided to two-sided loop extrusion in some molecules in vivo. However, our analysis suggests that two-sided extrusion should

be tightly coupled to stable chromatin binding. In particular, two-sided extrusion by stably bound condensin II is sufficient to linearly compact mitotic chromosomes in simulations. We thus hypothesize that condensins, particularly condensin II, may bind chromosomes as a dimer of condensin complexes in vivo. Such a dimer might have a longer residence time due to a larger protein-DNA interface, while also performing two-sided loop extrusion via its two protein motors.

Further single-molecule, biochemical, and structural studies could also help to understand how kleisins, Hawks, and post-translational modifications might generate diverse SMC complex symmetries and dynamics, and thus, functions. Such experiments, together with the principles established by our models, could clarify how the molecular properties of loop-extruding SMC complexes compact and organize chromosomes throughout the cell cycle.

## Materials and methods

### Simulations

Stochastic simulations of LEFs on a chromatin fiber are performed as previously described (*Goloborodko et al., 2016b*; *Banigan et al., 2020*) with adaptations as described in the Model section. The chromatin fiber is a one-dimensional lattice of length $L = 60000$ sites, each of which is taken to be $a = 0.5$ kb. LEFs bind to chromatin at rate $k_b$, and the two LEF subunits initially occupy two adjacent lattice sites upon binding the fiber. Each active subunit of the LEF may translocate. Translocation occurs in a directed manner away from the sites originally occupied by the LEF unless otherwise noted. Simulations typically consist of $N = 1000 - 2000$ LEFs (i.e. $d = L/N = 30 - 60$ lattice sites or $d = 15 - 30$ kb). Each simulation is run for a duration of $t_{\text{total}} = 400\tau_{\text{longest}}$, where $\tau_{\text{longest}}$ is the longest mean residence time. For each simulation, 100 data points are collected long after achieving steady state (which occurs after $t \approx 10\tau_{\text{longest}}$) from the time interval $300\tau_{\text{longest}} \leq t \leq 400\tau_{\text{longest}}$. Each simulation is run at least twice. The fraction, $f$, of the fiber compacted into loops is thus measured for each set of parameters with a standard error that is <5% of the mean. The simulation code is publicly available at (https://github.com/mirnylab/one_sided_extrusion/tree/master/mitotic/ ; *Banigan, 2020*; copy archived at swh:1:rev:b27012e95d354e8deaac5bcfdcb3c36b375626ce).

### Estimation of physiological values of $\lambda/d$

The physiological range of the ratio of the processivity to the mean separation is estimated as $10 < \lambda/d < 1000$, as calculated previously (*Banigan et al., 2020*). Processivity, $\lambda$, was estimated from experimental measurements of condensin's extrusion speed of $\sim 1$ kb/s in vitro (*Ganji et al., 2018*; *Golfier et al., 2020*; *Kong et al., 2020*) and measured and estimated residence times of order 1–100 min in vivo and in vitro minutes (*Gerlich et al., 2006a*; *Terakawa et al., 2017*; *Walther et al., 2018*; *Gibcus et al., 2018*). Mean separation, $d$, was determined by measured linear densities of 1 per 10–100 kb in vivo (*Takemoto et al., 2004*; *Fukui and Uchiyama, 2007*; *Walther et al., 2018*).

### Analysis of asymmetric extrusion

For the general estimate of the physiological range of asymmetries, the expected range of asymmetries is calculated from the symmetry scores measured by *Golfier et al., 2020*. The symmetry score is given by $S = (v_{\text{fast}} - v_{\text{slow}})/(v_{\text{fast}} + v_{\text{slow}})$, where $S = 0$ indicates perfectly symmetric two-sided extrusion and $S = 1$ indicates (completely asymmetric) one-sided extrusion. Using experimentally measured symmetry scores, we calculate that $v_{\text{slow}} > 0.025v_{\text{fast}}$ for two thirds of condensin loop extrusion events (green box in *Figure 1c*).

We also considered three scenarios of asymmetric extrusion to model experiments by *Golfier et al., 2020*. In the first scenario, all LEFs perform asymmetric two-sided extrusion, but there is a distribution of asymmetries; half of the LEFs have symmetry scores, $S$, uniformly randomly selected from the interval $[0, 0.8]$ and half of LEFs have $S$ randomly selected from $[0.8, 1]$. In the second scenario, only half of the LEFs perform asymmetric two-sided extrusion with asymmetries in the interval $[0, 0.8]$; the other half of the population performs one-sided extrusion (*i.e.*, $S = 1$).

In the third scenario, we calculated loop-growing and loop-shrinking velocities for condensins from single-molecule experiments by *Golfier et al., 2020*. Ten trajectories from the experiments were smoothed with a Savitsky-Golay filter with a second order polynomial and a window of 63 frames as in the previous analysis by *Golfier et al., 2020*. For each condensin trajectory, we then

calculated the mean size of loop-growing and loop-shrinking steps and computed mean loop-growing and loop-shrinking speeds for each of the two sides. We simulated two-sided asymmetric extrusion with these ten sets of velocities, with each set of velocities assigned to one tenth of the LEFs.

## Acknowledgements

We thank Hugo Brandão, Jan Brugués, and Maxim Imakaev for helpful discussions and Job Dekker and John Marko for critically reading the manuscript. This work was supported by the NIH Center for 3D Structure and Physics of the Genome of the 4DN Consortium (U54DK107980), the NIH Physical Sciences-Oncology Center (U54CA193419), and NIH grant GM114190.

## Additional information

### Funding

| Funder | Grant reference number | Author |
| --- | --- | --- |
| National Institutes of Health | U54DK107980 | Edward J Banigan<br>Leonid A Mirny |
| National Institutes of Health | U54CA193419 | Edward J Banigan<br>Leonid A Mirny |
| National Institutes of Health | GM114190 | Edward J Banigan<br>Leonid A Mirny |

The funders had no role in study design, data collection and interpretation, or the decision to submit the work for publication.

### Author contributions

Edward J Banigan, Conceptualization, Software, Formal analysis, Investigation, Writing - original draft, Writing - review and editing; Leonid A Mirny, Conceptualization, Supervision, Funding acquisition, Writing - review and editing

### Author ORCIDs

Edward J Banigan (iD) https://orcid.org/0000-0001-5478-7425
Leonid A Mirny (iD) https://orcid.org/0000-0002-0785-5410

### Decision letter and Author response

Decision letter https://doi.org/10.7554/eLife.63528.sa1

## Additional files

### Supplementary files

• Transparent reporting form

### Data availability

Software used to perform simulations is publicly and freely available at https://github.com/mirnylab/one_sided_extrusion/tree/master/mitotic (copy archived at https://archive.softwareheritage.org/swh:1:rev:b27012e95d354e8deaac5bcfdcb3c36b375626ce/). Data analyzed from single-molecule experiments was previously published as part of Golfier et al. eLife 9:e53885 (2020).

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

# Appendix 1

## Theoretical analysis of mixtures of LEFs with different dynamics

### Mixtures of one- and two-sided LEFs

We theoretically analyzed the ability of mixtures of one- and two-sided LEFs with different mean speeds and residence times to linearly compact the chromatin fiber. To compute the maximum achievable compaction, we considered two scenarios beyond the case of uniform speeds and residence times considered previously (*Banigan and Mirny, 2019*): (1) the case in which two-sided LEFs are long-lived compared to one-sided LEFs (i.e. $\tau_2 \gg \tau_1$) and (2) the case in which two-sided LEFs are short-lived compared to one-sided LEFs (i.e. $\tau_2 \ll \tau_1$). We consider speeds $v_1$ and $v_2$ that are sufficiently fast for a LEF to extrude all nearby chromatin into a loop before the LEF unbinds (i.e. $\lambda/d \gg 1$).

In both scenarios, it is sufficient to calculate the fractions, $f_1$ and $f_2$, of the fiber that would be extruded, respectively, by the one-sided LEFs and the two-sided LEFs for their respective linear densities and to-be-determined effective processivities. This calculation allows us to compute the fraction of the fiber that remains in unlooped gaps:

$$f_{\text{gap}} = (1 - f_1)(1 - f_2). \tag{1}$$

The total fraction of the fiber that is compacted into loops is then given by:

$$f = 1 - f_{\text{gap}}, \tag{2}$$

while the fold linear compaction is:

$$\mathcal{FC} = \frac{1}{1 - f}. \tag{3}$$

From previous mean-field theoretical calculations, we know that one-sided LEFs linearly compact a fraction $f_1 \approx 0.895$ of the fiber that is accessible to them (leaving $1 - f_1 \approx 0.105$ in unlooped gaps) (*Banigan and Mirny, 2019*). The fraction, $f_2$, compacted by the two-sided LEFs is determined below.

For $\tau_2 \gg \tau_1$, two-sided LEFs extrude on a fiber loaded with transient barriers (the one-sided LEFs), while one-sided LEFs act to compact a fiber that has already been partially compacted (by the two-sided LEFs) (*Figure 2b*, left). To compute the fraction, $f_2$, compacted by the two-sided LEFs, we consider that each two-sided LEF extrudes processively at speed $v_2$, but periodically encounters one-sided LEFs that act as transient barriers. The barriers limit the mean speed of the two-sided LEFs. Since the barriers are separated by a mean distance of $d_1$, disappear in mean time $\tau_1$, the two-sided LEFs have an effective velocity of:

$$v_2^{\text{eff}} = \frac{d_1}{\tau_1}, \tag{4}$$

which is the average distance that a two-sided LEF travels between barriers over the mean lifetime that each barrier is present. The resulting ratio of effective processivity ($\lambda_2^{\text{eff}} = v_2^{\text{eff}} \tau_2$) to mean separation between two-sided LEFs is:

$$\frac{\lambda_2^{\text{eff}}}{d_2} = \frac{\tau_2 d_1}{\tau_1 d_2}. \tag{5}$$

This ratio is larger for a longer relative two-sided LEF lifetime (larger $\tau_2/\tau_1$) because the two-sided LEFs outlast and extrude past a greater number of transient barriers. The ratio also grows for increasing distances between barriers (larger $d_1$) because two-sided LEFs can freely translocate for greater distances along the fiber.

The fraction, $f_2$, of the fiber that is compacted by the two-sided LEFs is equivalent to the fraction, $f_2(\lambda_2^{\text{eff}}/d_2)$, of the fiber compacted in a system of two-sided LEFs with a single residence time at the processivity-to-separation ratio $\lambda_2^{\text{eff}}/d_2$. The fraction $f_2(\lambda_2^{\text{eff}}/d_2)$ is obtained by simulations of two-sided LEFs (*Goloborodko et al., 2016b*; *Banigan et al., 2020*).

In contrast to the scenario above, with $\tau_2 \ll \tau_1$, two-sided extrusion occurs on a chromatin fiber that has been partially compacted by one-sided extrusion. We approximate this condition as two-sided LEFs performing loop extrusion on a background of effectively permanent barriers (long-lived one-sided LEFs) (*Figure 2b*, right).

Again, to determine the total fraction, $f$, of the fiber that is linearly compacted into loops, we first compute the effective velocity of the two-sided LEFs. The two-sided LEFs translocate until encountering barriers, spaced $d_1$ apart, after which they are permanently stalled (because the barriers are long-lived compared to the residence time of a two-sided LEF). Therefore, the effective velocity of a two-sided LEF is:

$$v_2^{\text{eff}} = \frac{d_1}{\tau_2}. \tag{6}$$

Now the ratio of effective processivity to separation is:

$$\frac{\lambda_2^{\text{eff}}}{d_2} = \frac{d_1}{d_2}. \tag{7}$$

This ratio is independent of the speed and residence times of the two-sided LEFs, because the one-sided LEFs appear to be permanent barriers to the two-sided LEFs, irrespective of $v_2$ and $\tau_2$ (provided that $\tau_2 \ll \tau_1$). Once again, the fraction of the fiber compacted by the one-sided LEFs is given by the fraction $f_2(\lambda_2^{\text{eff}}/d_2)$, obtained from simulations of two-sided LEFs.

Combining the above results, we find that the total fraction of the fiber that is compacted into loops is:

$$f = 1 - 0.105(1 - f_2(\frac{\lambda_2^{\text{eff}}}{d_2})), \tag{8}$$

where $\lambda_2^{\text{eff}} = v_2^{\text{eff}}\tau_2$ is set by either *Equation 5 or 7*, depending on $\tau_2/\tau_1$. The fraction, $f$, of the fiber compacted (*Equation 8*) can be expressed in terms of the fraction, $\phi_1$, of one-sided LEFs in the mixture by noting that $d_1/d_2 = (1 - \phi_1)/\phi_1$. Then, the ratio of the effective processivity to the mean separation is given by:

$$\frac{\lambda_2^{\text{eff}}}{d_2} = \begin{cases} \dfrac{\tau_2}{\tau_1}\dfrac{1-\phi_1}{\phi_1} & \text{for } \tau_2 \gg \tau_1, \\[2ex] \dfrac{1-\phi_1}{\phi_1} & \text{for } \tau_2 \ll \tau_1. \end{cases} \tag{9}$$

We interpolate between these two scenarios using the result from the previously developed mean-field theory (for $\tau_2 = \tau_1$, $f = (1 - (\phi_1/2)^2 + \ln(4/(\phi_1)^2))/(1 + \ln(4/(\phi_1)^2)))$ (*Banigan and Mirny, 2019*).

In summary, we expect the maximum achievable linear compaction to grow rapidly with increasing $\tau_2/\tau_1$ for mixtures with relatively long-lived two-sided LEFs. Compaction can be depressed by large fractions of one-sided LEFs. For mixtures with short-lived two-sided LEFs, compaction is insensitive to $\tau_2/\tau_1$, but it decreases as the fraction, $\phi_1$, of one-sided LEFs is increased. Interestingly, the theory predicts that the ratio of relative velocities, $v_2/v_1$, does not alter the maximum achievable linear compaction. The theoretical results are shown in *Figure 2d*, *Figure 2—figure supplement 1*, and *Figure 2—figure supplement 2*, where they are compared to the simulation results.

## Systems with two populations of one-sided LEFs with different dynamics
### Computing compaction from the effective processivities

We adapted the theory for mixtures of one- and two-sided LEFs to describe systems with one-sided LEFs with two different residence times, $\tau_S$ and $\tau_L$, which are short and long, respectively. Similar to *Equation 1*, the total fraction of the fiber that remains in unlooped gaps is:

$$f_{\text{gap}} = (1 - f_L)(1 - f_S), \tag{10}$$

where $f_L$ and $f_S$ are the fractions of the fiber that would be compacted by, respectively, the long-

lived and short-lived LEFs at their respective linear densities and to-be-determined effective processivities.

As before, short-lived LEFs act as barriers to long-lived LEFs, and thus limit the processivity of the long-lived LEFs:

$$\lambda_{\mathrm{L}}^{\mathrm{eff}} = v_{\mathrm{L}}^{\mathrm{eff}} \tau_{\mathrm{L}} = \frac{d_{\mathrm{S}}}{\tau_{\mathrm{S}}} \tau_{\mathrm{L}}. \tag{11}$$

Therefore, the fraction of fiber that the long-lived LEFs may compact is the fraction, $f_1$, compacted by one-sided LEFs at $\lambda_{\mathrm{L}}^{\mathrm{eff}}/d_{\mathrm{L}}$. That is given by $f_{\mathrm{L}} = f_1(\frac{\tau_{\mathrm{L}} d_{\mathrm{S}}}{\tau_{\mathrm{S}} d_{\mathrm{L}}})$, which is obtained from simulations of one-sided LEFs (*Banigan and Mirny, 2019*; *Banigan et al., 2020*).

The fraction, $f_{\mathrm{S}}$, compacted by the short-lived one-sided LEFs can be computed analogously to the calculation of compaction by short-lived two-sided LEFs in the previous section. Long-lived one-sided LEFs are effectively permanent barriers to short-lived one-sided LEFs, which have an effective processivity:

$$\lambda_{\mathrm{S}}^{\mathrm{eff}} = d_{\mathrm{L}}. \tag{12}$$

Then, the amount compacted by the short-lived LEFs is $f_{\mathrm{S}} = f_1(\frac{d_{\mathrm{L}}}{d_{\mathrm{S}}})$.

Combining the expressions for $f_{\mathrm{S}}$ and $f_{\mathrm{L}}$ and noting that $d_{\mathrm{S}}/d_{\mathrm{L}} = \phi_{\mathrm{L}}/(1 - \phi_{\mathrm{L}})$ (where $\phi_{\mathrm{L}}$ is the fraction of long-lived LEFs), we have:

$$f_{\mathrm{gap}} = (1 - f_1(\frac{\tau_{\mathrm{L}}}{\tau_{\mathrm{S}}} \frac{\phi_{\mathrm{L}}}{1 - \phi_{\mathrm{L}}}))(1 - f_1(\frac{1 - \phi_{\mathrm{L}}}{\phi_{\mathrm{L}}})). \tag{13}$$

As usual, the resulting fold compaction is given by $\mathcal{FC} = 1/(1 - f) = 1/f_{\mathrm{gap}}$; thus, minimizing $f_{\mathrm{gap}}$ in *Equation 13* maximizes fold compaction, $\mathcal{FC}$. *Equation 13* shows the dependence of the maximum fold compaction, $\mathcal{FC}_{\mathrm{max}}$, on several physical variables. The predictions of *Equation 13* are shown as a function of $\tau_{\mathrm{L}}/\tau_{\mathrm{S}}$ for several values of $\phi_{\mathrm{L}}$ in *Figure 2—figure supplement 4b*.

As in the case of one- and two-sided mixtures, the theory predicts that the maximum linear compaction by two populations of one-sided LEFs is independent of their relative velocities. The velocities considered are large enough to close gaps between neighboring LEFs (if properly oriented), and therefore, altering the velocities does not change the rates of gap formation and closure (*Figure 2—figure supplement 4d*).

As the ratio of residence times, $\tau_{\mathrm{L}}/\tau_{\mathrm{S}}$, increases, the fraction, $f_{\mathrm{gap}}$, of the fiber in gaps decreases. Increasing $\tau_{\mathrm{L}}/\tau_{\mathrm{S}}$ increases the effective processivity, $\lambda_{\mathrm{L}}^{\mathrm{eff}}$ (*Equation 11*). This effect increases the fraction, $f_{\mathrm{L}}$, of the fiber compacted by the long-lived LEFs, and consequently, the total fraction, $f$, of the fiber that is compacted.

Interestingly, $\mathcal{FC}_{\mathrm{max}}$ increases as the fraction, $\phi_{\mathrm{L}}$, of long-lived one-sided LEFs *decreases*. This contrasts with mixtures of one- and two-sided LEFs, for which compaction increases as the fraction of long-lived two-sided LEFs increases (*Equations 8 and 9*). Decreasing $\phi_{\mathrm{L}}$ increases the distance, $\lambda_{\mathrm{S}}^{\mathrm{eff}}$, that the short-lived LEFs may travel (i.e. the processivity, *Equation 12*), which increases their compaction ability. Moreover, while decreasing $\phi_{\mathrm{L}}$ may decrease $\lambda_{\mathrm{L}}^{\mathrm{eff}}$, this decrease can be offset by increasing the ratio of residence times, $\tau_{\mathrm{L}}/\tau_{\mathrm{S}}$ (which increases $\lambda_{\mathrm{L}}^{\mathrm{eff}}$; *Equation 13*). Altogether, in the limit of very small but nonzero $\phi_{\mathrm{L}}$ and very large $\tau_{\mathrm{L}}/\tau_{\mathrm{S}}$, the maximum fold compaction is $\mathcal{FC}_{\mathrm{max}} \approx 90$ (each population of LEFs compacts ~9.5-fold).

## Computing compaction by counting unlooped gaps

To better understand the system with two types of one-sided LEFs, we developed an alternative theory based on counting the number of unlooped gaps between loops in the limit of a very large disparity between LEF residence times (very large $\tau_{\mathrm{L}}/\tau_{\mathrm{S}}$). By counting the number of gaps and estimating their sizes by the mean separation between LEFs, we determine the fraction of the fiber that is not compacted into loops; using *Equations 2 and 3*, we compute the fold compaction.

First, because long-lived LEFs remain on the chromatin fiber longer than the transient barriers (the short-lived LEFs), they can achieve their maximal ~10-fold compaction when their processivity-to-density ratio, $\lambda_{\mathrm{L}}/d_{\mathrm{L}}$, is sufficiently large. We can thus use the previously developed mean field

theory of one-sided LEFs (*Banigan and Mirny, 2019*) to count the number of gaps, $N_{g,\mathrm{L}}$, between long-lived LEFs:

$$N_{g,\mathrm{L}} = \frac{N_{p,\mathrm{L}}}{4}, \tag{14}$$

where $N_{p,\mathrm{L}}$ is the number of long-lived LEFs that form ''parent'' loops (as opposed to being nested as ''child'' loops). Furthermore, the previous mean field theory shows that:

$$N_{p,\mathrm{L}} = \frac{N_{\mathrm{L}}}{1 + \ln 4}, \tag{15}$$

where $N_{\mathrm{L}}$ is the total number of long-lived LEFs.

The long-lived gaps are partially extruded into loops formed by short-lived LEFs. Similar to above, we expect the number of short-lived gaps, $N_{g,\mathrm{S}}$, to be related to the number of short-lived parent loops, $N_{p,\mathrm{S}}$, by:

$$N_{g,\mathrm{S}} = \frac{N_{p,\mathrm{S}}}{4}. \tag{16}$$

However, we expect that only a small fraction of the short-lived LEFs will reside within the long-lived gaps and form parent loops, depending on the fraction, $f_{\mathrm{L}}$, compacted by the long-lived gaps. Therefore, the number of short-lived parent loops is:

$$N_{p,\mathrm{S}} = (1 - f_{\mathrm{L}}) \frac{N_{\mathrm{S}}}{1 + \ln 4}. \tag{17}$$

As previously calculated (*Banigan and Mirny, 2019*), $f_{\mathrm{L}} = (3 + 4\ln 4)/(4 + 4\ln 4) = 0.895$. The gaps between short-lived LEFs are not extruded into loops, and they have size $g_{\mathrm{S}} = d_{\mathrm{S}}$.

Finally, we account for the unlooped gaps between long-lived LEFs and short-lived LEFs. We make the mean-field assumption that these gaps occur at the each of the boundaries of the long-lived gaps with 50% probability and have size $g_b = d$, where $d = L/(N_{\mathrm{L}} + N_{\mathrm{S}})$, where $L$ is the length of the chromatin fiber. These boundary gaps occur, on average, once per long-lived gap, or equivalently, once per four long-lived loops:

$$N_{g,\mathrm{b}} = \frac{N_{p,\mathrm{L}}}{4}. \tag{18}$$

The total fraction of the chromatin fiber that is not compacted into loops is then given by:

$$f_{\mathrm{gap}} = \frac{N_{g,\mathrm{S}} d_{\mathrm{S}} + N_{g,\mathrm{b}} d}{L} \tag{19}$$

Combining *Equations 14–19*, the total fraction of the fiber compacted into loops is:

$$f = 1 - f_{\mathrm{gap}} = \frac{16(1 + \ln 4)^2 - (1 + 4\phi_{\mathrm{L}}(1 + \ln 4))}{16(1 + \ln 4)^2}, \tag{20}$$

where we have used the fact that the numbers of long- and short-lived LEFs can be written in terms of the fraction, $\phi_{\mathrm{L}}$, of long-lived LEFs as $N_{\mathrm{L}} = \phi_{\mathrm{L}}(N_{\mathrm{L}} + N_{\mathrm{S}})$ and $N_{\mathrm{S}} = (1 - \phi_{\mathrm{L}})(N_{\mathrm{L}} + N_{\mathrm{S}})$, respectively.

We may now determine the degree of compaction that can be achieved as a function of the fraction of long-lived LEFs, $\phi_{\mathrm{L}}$; the prediction is shown by the black curve in *Figure 2—figure supplement 4c*. In the limit of $\phi_{\mathrm{L}} \to 0$, we find:

$$f = \frac{16(1 + \ln 4)^2 - 1}{16(1 + \ln 4)^2} = 0.989, \tag{21}$$

or 91-fold compaction, close to the theoretical prediction in the previous section. In the limit of $\phi_{\mathrm{L}} \to 1$, we find:

$$f = \frac{11 + (4\ln 4)(7 + 4\ln 4)}{16(1 + \ln 4)^2} = 0.884, \tag{22}$$

or 8.6-fold compaction. This is close, but not precisely equal, to the 9.5-fold compaction ($f = 0.895$) for a single population of LEFs calculated previously (*Banigan and Mirny, 2019*). This discrepancy can be remedied by more carefully counting the boundary gaps, as described in the following subsection.

## Refined counting of boundary gaps

The mean-field counting of the number of unlooped gaps (*Equations 16 and 18*) fails in two ways if the number of short-lived parent loops is comparable to or smaller than the number of long-lived gaps.

First, in the above argument, we made a simplifying assumption in counting the number of boundary gaps between short-lived and long-lived parent loops (*Equation 18*). We assumed that the number of boundary gaps equals the number of gaps between long-lived parent loops, $N_{g,b} = N_{g,L}$; this assumption is clearly violated if the number of short-lived parent loops falls below the number of long-lived gaps ($N_{p,S} < N_{g,L}$), which occurs for $\phi_L > (2 + \ln 4)^{-1} = 0.295$ in the above theory. This issue is corrected by:

$$N_{g,b} = \min(N_{g,L}, N_{p,S}). \tag{23}$$

Second, we assumed that $N_{g,S} = N_{p,S}/4$ holds exactly, even though it clearly must be violated when there are fewer than two short-lived parent loops in a long-lived gap ($N_{p,S} < 2N_{g,L}$), because a short-lived gap cannot be formed. This assumption is violated for $\phi_L > (3 + 2\ln 4)^{-1} = 0.173$. We correct this issue by:

$$N_{g,S} = \begin{cases} \frac{N_{p,S} - N_{g,L}}{4} & N_{p,S} > N_{g,L} \\ 0 & \text{else.} \end{cases} \tag{24}$$

Altogether, we now find that the fraction of the fiber that remains uncompacted in gaps is:

$$f_{\text{gap}} = \frac{N_{g,S}d_S + N_{g,b}d + (N_{g,L} - N_{g,b})d_L}{L}. \tag{25}$$

After simplification, the fraction compacted is:

$$f = 1 - f_{\text{gap}} = \begin{cases} \frac{1 + 16(1-\phi_L)(1+\ln 4)^2 - \phi_L(2 - 4\phi_L(1+\ln 4) - 3\ln 4)}{16(1-\phi_L)(1+\ln 4)^2}, & \phi_L \leq \frac{1}{2+\ln 4} \\ \frac{1 + \phi_L + \phi_L^2 + (\phi_L \ln 4)(7 + 4\ln 4)}{4\phi_L(1+\ln 4)^2}, & \phi_L > \frac{1}{2+\ln 4}. \end{cases} \tag{26}$$

As shown by the gray curve in *Figure 2—figure supplement 4c*, this modified theory has similar behavior to the mean-field theory in the previous section. This refined counting argument better estimates the ratio of the total number of gaps, $N_{g,L} + N_{g,S}$, to total number of parent loops, $N_\ell = N_{p,L} + N_{p,S}$ (inset to *Figure 2—figure supplement 4c*). In addition, at $\phi_L = 1$, this theory precisely reproduces the mean-field theory for a single population of one-sided LEFs (*Banigan and Mirny, 2019*).

# Appendix 2

## Simulations of other types of mixtures of LEFs

### Compaction by a mixture of LEFs does not depend on relative velocities

We speculated that rapid extrusion by two-sided LEFs ($v_2 > v_1$) might generate large two-sided loops, leading to nesting by the one-sided LEFs that form smaller loops; consequently, there might be fewer gaps formed by divergently oriented ($\leftarrow\rightarrow$) LEFs. Consistent with this idea, previous modeling with two varieties of two-sided LEF showed that LEFs with large processivities ($\lambda = v/k$) formed loops that were split by the extruded loops of LEFs with small processivities (*Gibcus et al., 2018*). On the other hand, theory (see Appendix 1) predicts that the relative extrusion velocities, set by the ratio $v_2/v_1$, do not set the maximum compaction. To test the theoretical predictions for linear compaction against the possibility of forming a hierarchy of loops, we simulated mixtures of one- and two-sided LEFs with different velocities, but identical mean residence times.

We find that the maximum achievable linear compaction is insensitive to the relative velocities, $v_2/v_1$, as predicted by the theory (*Figure 2—figure supplement 1a*). This is consistent with the previous theoretical model (*Banigan and Mirny, 2019*), which assumed rapid extrusion and closure of all possible gaps between LEFs in steady state. Altering the velocities of the LEFs does not alter the steady-state rate at which LEFs unbind from and rebind to the chromatin fiber; consequently, unlooped gaps are not eliminated any more effectively than in the scenario with a single mean velocity (*Figure 2—figure supplement 1b*). As a result, the maximum achievable fold compaction, $\mathcal{FC}_{\mathrm{max}}$, for each fraction, $\phi_1$, of one-sided LEFs is given by mean-field theoretical limit calculated for mixtures of LEFs with uniform velocities (*Banigan and Mirny, 2019*; *Figure 2—figure supplement 1a and c*). Thus, irrespective of the relative velocities of one- and two-sided LEFs, a minimum fraction of two-sided LEFs of $\phi_2 \approx 0.84$ LEFs is required to achieve 1000-fold linear compaction.

### Compaction by one-sided LEFs is ineffective even with a long-lived subpopulation

We explored whether having a two populations with different mean residence times would enhance linear compaction for a chromosome with only pure one-sided LEFs. As described in Appendix 1, we adapted the theory for mixtures of one- and two-sided LEFs to systems with two populations of one-sided LEFs, each with different residence times, $\tau_{\mathrm{S}}$ and $\tau_{\mathrm{L}}$ (denoting short and long). Once again, compaction is independent of the relative velocities of the two populations (*Figure 2—figure supplement 4a*). The theory predicts and simulations show that increasing the ratio of residence times, $\tau_{\mathrm{L}}/\tau_{\mathrm{S}}$, increases the maximum fold compaction, $\mathcal{FC}_{\mathrm{max}}$ beyond the ~10-fold limit for one-sided LEFs with single mean residence time ($\tau_{\mathrm{L}}/\tau_{\mathrm{S}} = 1$) (*Figure 2—figure supplement 4b and c*). Interestingly, the compaction increases as the fraction of long-lived LEFs, $\phi_{\mathrm{L}}$, is decreased. This effect occurs because decreasing the number of long-lived large loops increases the effective processivity of the short-lived LEFs (alternatively, the relationship can be understood by counting gaps and loops; see Appendix 1). Nevertheless, we find that linear fold compaction is still limited to a maximum of $\mathcal{FC}_{\mathrm{max}} \approx 90$ (i.e. 9.5-fold compaction from each population of one-sided LEFs).

