## [Decision Letter]

Congratulations, we are pleased to inform you that your article, "The interplay between asymmetric and symmetric DNA loop extrusion", has been accepted for publication in *eLife*.

Reviewer #2:

Banigan and Mirny revisit their modeling approach previously published in *eLife* (Banigan et al., 2020) to incorporate new experimental findings that in vivo condensins likely utilize asymmetric two-sided loop extrusion in addition to the one-sided and symmetrical two-sided extrusion examined in the previous work. By adding variable residence time and asymmetric extrusion, the authors are able to model physiologically relevant compaction consistent with experimentally determined levels and properties of Condensins I and II. These findings represent a significant advancement over the previous work, the conclusions are well-justified, and a well-written and conceived Discussion gives a valuable perspective of the current state of the field. As a result, I support the publication of this manuscript in *eLife* without additional data.

Reviewer #3:

In this paper, Banigan and Mirny study the effects of different loop-extrusion symmetries to the overall compaction of chromatin. This study is motivated by recent work showing that condensin loop extrusion is not one-sided as shown purely in vitro, but rather displays a mixed population of symmetries. From a theory point of view, one-sided loop extrusion cannot condense DNA sufficiently, so since the first in vitro demonstration of one-sided loop extrusion it has not been clear how to marry theory and experiments with the requirements of DNA condensation in cells. Based on the results from *Xenopus laevis* egg extracts, the authors extend previous models of loop extrusion by allowing populations of loop extruders with different residence times, and active loop extruding subunits that translocate at the different speed. They show that the distribution of symmetries observed in these experiments (even though most of loop extrusion is one-sided) can account for over 1000 fold condensation (as in cells). They also considered the effects of two condensins (I and II) corresponding to purely one and two sided extrusion. Interestingly they show that a longer residence time of a two-sided condensin would also account for over 1000 fold condensation. As a consequence, their model suggests a testable prediction for the activities of condensin I and II. Overall I found the paper very clear and systematic, and their findings are an important contribution to understanding how loop extruding factors may contribute to the overall organization of chromatin, even though this work is an extension of their previous models. As such I think it fits as a research advances and support publication as is.